# Gait Apraxia and Hakim’s Disease: A Historical Review

**DOI:** 10.3390/biomedicines11041086

**Published:** 2023-04-03

**Authors:** David Milletti, Filippo Tamburini Randi, Giuseppe Lanzino, Fernando Hakim, Giorgio Palandri

**Affiliations:** 1Unit of Rehabilitation Medicine, IRCCS Istituto delle Scienze Neurologiche di Bologna, 40139 Bologna, Italy; 2Unit of Neurosurgery, IRCCS Istituto delle Scienze Neurologiche di Bologna, 40139 Bologna, Italy; 3Department of Neurosurgery, Mayo Clinic, Rochester, MN 55905, USA; 4Department of Neurosurgery, Hospital Universitario Fundación Santa Fé de Bogotá, Bogotá 220246, Cundinamarca, Colombia

**Keywords:** gait apraxia, frontal gait, Bruns apraxia, idiopathic normal pressure hydrocephalus, Hakim’s syndrome

## Abstract

In 1965, Prof. Salomón Hakim described, for the first time, a condition characterized by normal pressure hydrocephalus and gait alterations. During the following decades, definitions such as “Frontal Gait”, “Bruns’ Ataxia” and “Gait Apraxia” have been frequently used in pertinent literature in the attempt to best define this peculiar motor disturbance. More recently, gait analysis has further shed light on the typical spatiotemporal gait alterations that characterize this neurological condition, but a clear and shared definition of this motor condition is still lacking. In this historical review, we described the origins of the terms “Gait Apraxia”, “Frontal Gait” and “Bruns’ Ataxia”, starting with the first works of Carl Maria Finkelburg, Fritsch and Hitzig and Steinthal during the second half of the 19th century and ending with Hakim’s studies and his formal definition of idiopathic normal pressure hydrocephalus (iNPH). In the second part of the review, we analyze how and why these definitions of gait have been associated with Hakim’s disease in the literature from 1965 to the present day. The definition of “Gait and Postural Transition Apraxia” is then proposed, but fundamental questions about the nature and mechanisms underlying this condition remain unanswered.

## 1. Introduction

*“It is obviously easier to say what the gait disturbance is not than what it is”* [1].

In recent years, gait analysis (GA) has progressively led to a renewed interest among researchers and physicians in gait alterations of idiopathic normal pressure hydrocephalus (iNPH) [2,3,4,5]. In particular, wearable inertial sensors have recently allowed the analysis of spatiotemporal data that define the main stereotyped features of this gait disturbance [6,7]. On the other hand, data on the origin and pathophysiology of this gait alteration are still lacking. Definitions such as “Frontal Gait”, “Bruns’ Ataxia” and “Apraxia of Gait”, frequently used in pertinent literature to describe iNPH gait, have very little consensus between experts. Given the lack of consensus about the use of the definition of gait apraxia in Hakim’s disease, our review chronologically analyzes the evolution of gait apraxia’s concept in both neurological and gait analysis literature. The aim of the review was then to describe when, how and why iNPH gait abnormalities started to be described as “Apraxia of Gait”. The secondary aims of this narrative review were to highlight the limitations of a description of the sole gait alterations. Based on these limitations, we further proposed a new definition of “Gait and Postural Transitions Apraxia”.

We divided this review into two parts, before and after 1965, considering as a milestone and watershed the publication in which Prof. Salomón Hakim described, for the first time, three cases of normal pressure hydrocephalus [8]. The MeSH (Medical Subject Headings) vocabulary has been used to define appropriate keywords for the research. The terms “normal pressure hydrocephalus”, “Hakim’s Syndrome”, “apraxia of gait”, “frontal gait”, “magnetic gait” and “Bruns Gait” were identified. Based on these keywords, the authors performed a search on PubMed® and Web of Science® databases, including case reports, case series, experimental studies and reviews. Only articles in English, French and German were included in the review. The first two authors reviewed all the selected articles in a chronological order, then discussed and classified, one by one, every study on the basis of these parameters: relevance to apraxia of gait, normal pressure hydrocephalus or to different kinds of hydrocephalus or frontal pathology and temporal localization (before or after 1965). At the end of this evaluation process, 32 publications were included in the narrative review (Table 1).

Two authors respectively wrote the first part (Tamburini Randi reviewed the articles written before 1965) and the second part (Milletti reviewed research published after 1965) of the manuscript. A step-by-step review of the manuscript’s work in progress was then performed, providing harmonization of the contents in a coherent conceptual framework. Collegial revision and discussion was performed by all authors to finalize the manuscript.

## 2. The Definition of Apraxia and First Observations of Hydrocephalus Gait Alterations (1870–1892)

In 1870, the German psychiatrist Carl Maria Finkelburg, observing some motor disturbances in aphasic persons, described a loss of dexterity in executing simple actions; he called this condition “motor asymbolia” [9]. This theory was met with great approval by his mentor Wernicke, who hypothesized damage in frontopontocerebellar fibers as the cause of such disturbances [10]. In the same period, Fritsch and Hitzig, based on electrophysiological experiments conducted on animals, stated that motor functions of the brain are not simply dependent on the unique connections of cortical neurons to muscle fibers; any action needs a representative image of a specific sequence of muscular actions in the brain in order to accomplish a finalistic movement. Their theory gave birth to the concept of “cortical images”, reciprocal connections whose purpose is to integrate different functions from different areas [11]. In 1871, Steinthal, in an attempt to describe a motor disturbance strictly associated to aphasia, used the definition of “apraxia” for the first time (“Diese Apraxie ist eine offenbare Steigerung der Aphasie”) [12]. The term “Apraxia” derives from the word “πρᾶξις”, which indicates the execution of an action from its purpose to its final result (the “operative action”), with a “privative α” at the beginning. While these pioneering scientists were starting to describe and define the nature of apraxia in the second half of the 19th century, someone else was taking the first fundamental steps in the study of gait alterations in hydrocephalus and frontal pathology. In an article dated 1865, Samuel Wilks described some patients affected by chronic adult hydrocephalus (probably an obstructive form). Their gait was “stooped and feeble”, but no more details were given whatsoever. To the best of our knowledge, this is the first paper in the modern era reporting an alteration of gait associated with hydrocephalus. The author described it as a different form of ataxia “of no cerebellar origin” [13]. In 1892, Bruns reported an interesting case series of gait alterations in patients affected by frontal lobe tumors or hematomas invading frontal regions [14]. Bruns defines it as follows: “the gait is ataxic, (the patient) skids, sometimes one footsteps over the other. When standing, must be held: if you let go of the patient, he falls with the inclination to the right and backwards: it stands broadly”. This gait pattern was distinguished from cerebellar ataxia by retropulsion and clumsy leg movement without a normal stepping motion. Furthermore, typical signs of cerebellar disease were not present. Discussing the possible causes of these signs, Bruns mentioned Wernicke’s theory of frontopontocerebellar tract damage and believed that “frontal ataxia” was due to paralysis of the muscles of the trunk, which he believed to be controlled by the frontal lobe. In particular, Bruns supposed that ataxia in frontal lobe diseases were due to compression of the cerebellum resulting from the displacement of the brain, thus interfering with the frontopontocerebellar tract. In the following decades, the term “frontal gait” or “Bruns’ gait” would be frequently used to describe typical hydrocephalus-associated gait.

## 3. From Liepmann’s Classification to the Definition of Apraxia of Gait: Sixty Years of Observations and Theories (1900–1965)

Following the main theory of reciprocal connections of cortical neurons of different areas [11], at the very beginning of the 20th century, the German neurologist Hugo Karl Liepmann [15] introduced a new entity to define some specific conditions affecting the central nervous system. In his “*Das Krankheitsbild der Apraxie*” [16], the term “apraxia” identified a peculiar dissociation of the concept of movement from the motor function, resulting in some specific abnormal postural and motor behaviors. Liepmann described apraxia as a motor-asymbolia derived from the loss of some specific cortical connections accounting for the “image” (the symbol) of motor function. In particular, he defined “limb-kinetic apraxia” as an impossibility to activate the correct sequence of cortical motor neurons to achieve a fluid movement, caused by a disturbance of kinaesthetic images of limbs. Liepmann identified the origin of this kind of apraxia in the disruption of cortical and subcortical areas of the frontal lobe, without pointing to a precise “area of frontal apraxia”. In line with Liepmann’s theories, the German neurologist Fritz Hartmann (1907) introduced the terms “trunk apraxia” and “lower limb apraxia” to describe characteristic altered motor patterns in patients affected by frontal lobe diseases [17]. Gerstmann and Schilder (1926), reporting two cases of patients affected by frontal lesions (abscess and encephalitis), introduced the concept of “apraxia of gait” for the first time [18]. The key aspects of this form of apraxia were defined as follows: “the patient is unable to use both legs to walk even when supported; he cannot lift them from the floor. There’s a high frequency of losses of balance and falls”. Van Bogaert and Martin (1929), strictly referring to the studies conducted by Gerstmann and Schilder, described the condition of “rench de la marche” in a woman affected by frontal lobe abscess, characterized by a peculiar aspect: while lying down on the bed, all the simple and monosegmental actions of lower limbs were defined as perfectly executable but her walking ability was completely impaired; the hips, ankles and knees were described rigid and limiting the ability to elevate feet from the floor; the gait was slow, shuffling, hesitant and characterized by short steps; and static balance was compromised and falls were frequent [19]. In 1936, a first attempt to operate a synthesis of previous observations and hypotheses was made by French psychiatrist and neurologist Paul Delmas Marsalet, who published “Lobe frontal et équilibre” [20], in which he proposed three types of topographic syndromes. In his theory, cortical damage causes pseudo-cerebellar symptoms (especially ataxia and Bruns ataxia), while white matter damage determines a loss of trunk muscular tone, resulting in astasia and “apraxia of the trunk” (following Hartmann’s theory). A more diffuse lesion of cortical and subcortical areas in the frontal lobe can determine a complex apraxic syndrome with both gait and postural symptoms. However, the author did not exclude the possibility of a complex integration of balance and gait functions all over the cortical gray matter of the frontal lobe, and a specific area responsible for all the syndromes he observed could not be found. Wernicke’s theory of frontopontocerebellar fiber damage was still considered by the author as a possible cause of those symptoms, although it was not considered sufficient to explain them all. A few years later, the scientific community started to focus on the relationship between gait alterations in frontal pathology and hydrocephalus. Yakovlev (1947), in his “Paraplegias of hydrocephalics; a clinical note and interpretation”, pointed out that the enlargement of frontal horns of lateral ventriculi could damage the adjacent frontopontine fibers, thus extending the role of subcortical frontopontocerebellar fibers in the regulation of trunk posture as well as in gait control, even for the condition of hydrocephalus [21]. So, still in the middle of the 20th century, Wernicke’s theory of frontocerebellar pathway alterations was the main accepted explanation of the relationship between gait and frontal pathology. In the following years, Danny-Brown proposed a different hypothesis [22]; in his “The nature of apraxia” (1958), he stressed the connection between apraxia, frontal lobe pathology and gait alterations, focusing on the typical shuffling, magnetic gait: “when the patient puts the affected foot on the floor, the lower limb stiffens, becomes glued to the floor so that steps are made only with great difficulty. Only when the heel is, at last, raised from the floor, can the foot be lifted, and a complete step made”. In his idea of a “slipping clutch syndrome”, frontal lobe pathologies are responsible for an instinctive grasp reaction, a type of kinetic limb apraxia. Danny-Brown proposed that this grasp reaction, determining upper and lower limb motor alterations, can also affect gait ability. In 1958, Meyer and Barron further stressed the gap between apraxia and ataxia in a case series of seven patients affected by frontal lobe diseases [23]. They defined “apraxia of gait” as the “loss of ability to properly use the lower limbs in the act of walking which cannot be accounted for by demonstrable sensory impairment or motor weakness”. The authors also noted: “in general, features that are characteristic of apraxia of the legs but which are absent in cerebellar disease are slowness in initiation of movement, tendency to deviate to the side of the apraxic limb in performing the “star-gait” test (because the affected limb takes shorter steps and the trunk tends to deviate to the affected side), perseveration of posture, rigidity, *gegenhalten*, hypokinesis and difficulty in performing such motions with the feet as drawing a circle, tapping the heel on the floor and kicking an imaginary ball”.

In 1965, Prof. Salomón Hakim (Figure 1), for the first time in medical history, described and defined the condition of Normal Pressure Hydrocephalus [24], characterized by urinary incontinence, impairment of cognitive functions and unsteadiness of gait [8,25].

The term ataxia (a derivation from Bruns ataxia) was used to define motor alterations, although the typical cerebellar symptoms could not be recognized; the Romberg test was negative, and gait was not similar to that associated with cerebellar diseases. The gait of three patients was presented as follows for the respective cases. Case 1: “In standing the patient was insecure”; “in walking tandem she occasionally lurched slightly to one or the other side”; “She walked slowly on a normal base, but the steps were shortened”; “She took a few short, shuffling steps with the left foot, bringing the right one up to it”. Case 2: “She walked with small, 12-inch steps, slightly stooped, and tended to lean forward. She could not walk a line heel to toe. The Romberg sign was absent, but she swayed more than usual”. Case 3: “difficulty in walking appeared: it was described as a shuffling stiffness, a tendency to lose balance and need to hold the railing of the stairs. Impairment of gait was variable and at its worse he could not walk at all. His posture was stooped and he tended to topple backward”; “Walked slowly in what appeared to be a “careless” way, staggering and lurching slightly on a variable base”; “He tended to scuff his left toe and was unbalanced when making a quick turn. On tandem gait he fell to the left. He could balance on the right foot with his eyes open, but not on the left”. In order to explain the origin of such gait disturbance, Hakim recalled the Yakovlev paper, in particular the theory of frontopontocerebellar tract damage; in their pneumoencephalographic tests, they noticed a more prominent enlargement of frontal horns of lateral ventricles; in his opinion, this aspect might strengthen the hypothesis of a progressive lesion in those descending fibers [3,17]. The historical evolution of the concept of gait apraxia in frontal diseases is summarized in Figure 2.

## 4. From Prof. Hakim’s Definition of Normal Pressure Hydrocephalus to the Present Day: Can We Really Define the Altered Gait in iNPH as “Apraxic”?

In two different case series published between 1977 and 1982, Fisher CM, who had previously co-authored with Prof Hakim in his seminal “Symptomatic occult hydrocephalus with normal cerebrospinal fluid pressure”, interesting data were presented about the clinical presentation of iNPH [1,26]. In his “The Clinical Picture in Occult Hydrocephalus” paper, Fisher stressed the importance of gait alterations as they very often represent the onset of iNPH and the most evident and worst symptom composing the triad (in half of the persons it was evaluated as problematic at the level of mental disturbance).

In his “Hydrocephalus as a cause of disturbances of gait in the elderly” paper (1982) [26], he addressed, more specifically, the gait abnormalities in 50 persons diagnosed with normal pressure hydrocephalus in over 20 years of practice. They are defined as follows: “poor balance, off balance, unsteady, wobbly, staggering, drunken, falling, and difficulty on stairs and curbs. Falling forward was mentioned in three patients, backwards in three, and to the side in one. Patients often required a cane, crutch, or walker”. Twelve persons even complained of lower limb tiredness and weakness in walking and standing. Fisher kept describing the evolution of those disturbances over time: “the steps became shorter and shuffling, scuffing occurred, and turning was slow, multistepped, and precarious. Walking unassisted gradually became unsafe, and in the advanced state, standing, sitting, and finally turning over in bed were impossible. Yet at this end stage of the process the legs functioned well when the patient was lying on the back, indicating that the gait disorder fell within the sphere of frontal gait apraxia”. In a 1981 case series involving around six persons affected by communicating (non-obstructive) hydrocephalus, Estañol focused on “perseveration” [27]: “There is an inability to shift from one movement to another although the motor task remains always clear”. Estañol considered this perseveration, induced by proprioceptive stimuli, to be a “motor-apraxia” (lower limb-specific apraxia), in which the ability to mentally draw the correct sequence of muscle actions necessary to execute a movement is preserved but this sequence cannot be translated to muscle action. He also wrote: “It should be stressed, at the outset, that apraxia of gait is not a form of ataxia but a defect at a higher level of neural organization”. We can find here an attempt to explain gait alteration in hydrocephalus with Danny-Brown’s model of kinetic apraxia [22]. Five years after Estañol’s paper, his idea of perseveration in gait seemed to find an indirect confirmation in a study by Evert Knutsson on patients affected by Normal Pressure Hydrocephalus [2]. Taking advantage of the techniques of modern gait analysis, Knutsson used intermittent light photography and EMG to describe the activation of different muscle groups both before and after the shunting procedure. He described a pattern of persistent contraction of antigravitational muscles throughout the gait cycle, in opposition to the normal phasic activity. A reduction in gait speed and in angular variation of all the major joints was present during the gait phase. Knutsson’s study has been followed, in the last three decades, by a large amount of gait analysis study of iNPH. However, many of these studies were predominantly focused on spatiotemporal parameters, their modifications after the Tap Test (TT) or shunt surgery, or differential diagnosis from iNPH-mimics [3,4,28,29]. The term “gait apraxia” is often absent in these studies, replaced by definitions such “gait alteration” or “frontal gait”. No further theories or new classifications about apraxia in iNPH emerged from gait analysis literature during these years.

## 5. Discussion

Idiopathic normal pressure hydrocephalus can be characterized, often in its early stage, by gait alteration in the absence of a demonstrable muscular weakness and cerebellar and sensory impairment. This condition, according to the definition of Meyer and Barron [23], could be defined as apraxia of gait. Apraxia of gait remains, however, a controversial entity. Two main parallel theories have developed after Liepmann’s formal definition of apraxia [16]. The first considers the frontal lobe as a fundamental regulator of gait function, thus giving credit to the existence of “apraxia of gait” [18,19,20]. In particular, recent studies claimed a possible role of a “disconnection” of the supplementary motor area (SMA) in the genesis of this motor disturbance [30,31]. The second derives from Bruns ataxia, in which frontopontocerebellar fiber damage can produce a pseudocerebellar syndrome, otherwise named “frontal ataxia” [14,21,25]. A connection between iNPH and apraxia according to the classic Liepmann model has not yet been demonstrated, although according to some authors it is possible to hypothesize a relationship with limb-kinetic apraxia in the presence of suffering of the frontal lobes and corpus callosum. This would be true assuming periventricular damage in iNPH, particularly at the level of the frontal horns [14,21,32]. As well as being controversial, the expression “apraxia of gait” probably also represents an incomplete definition of a more complex motor syndrome. Hartmann and Marsalet [17,20] already dealt with the role of postural symptoms in determining an impairment of gait and posture; more recently, Dale et al. proposed a model that differentiates a pure limb apraxia from conditions that affect gait and equilibrium [33]. Referring to a Geschwind analysis dating back to 1975 [34], the authors distinguished the neural connections responsible for the movement of limbs (whose hierarchically last neural connection is the corticospinal tract) from those that control static and dynamic equilibrium (which are found in tectospinal, vestibulospinal and uncrossed anterior pyramidal tracts, their last neural pathway). The authors proposed that many historical descriptions of the so-called gait apraxia or Bruns gait are actually characterized by a lack of stability during postural transitions and adjustments, thus configuring the new condition of “apraxia of postural transitions”. Dale et al. consider that many disturbances associated with frontal lobe diseases, even iNPH, are frequently associated with an instability of both the static and dynamic components of equilibrium altogether, with a difficulty in the progression from static to dynamic or vice versa. As well as being controversial and potentially incomplete, the definition of “Apraxia of Gait”, when we refer to iNPH, seems not completely appropriate. Despite the fact that the etymology of the word “apraxia” suggests the “inability to execute a precise motion (gait) with a blockage in the mental process responsible for it”, in the majority of cases, iNPH patients can instead walk independently, with or without help (supervision, slight assistance, aids, etc.), completely losing autonomy only in the late stage of the pathology. Rather than being unable to walk, people with iNPH appear unable to voluntarily modulate their gait. The gait pattern of iNPH patients is typically characterized by strongly stereotyped elements [33], including enlarged base, short walk and low stride height (shuffling gait), an increase in the double support time, a characteristic EMG pattern of contemporary contraction of antagonist muscles, difficulty in ascents and difficulty in directional changes. A large amount of gait analysis studies have confirmed these data over the years. These observations, interestingly, lead us back to 1981 and Estañol’s idea of “perseveration” [27]. Typical motor alterations in Hakim’s Disease are summarized in Figure 3. We suggest the new definition of "Gait and Postural Transition Apraxia" to emphasize the idea of a complex syndrome that can potentially affect multiple motor abilities. The sum of these conditions can affect a patient’s independence and determine fall risk (Figure 3). In the near future, implications for research could result from a more extensive definition of this motor syndrome. To date, many experimental studies about iNPH and shunt response have considered isolated gait analysis instead of evaluating gait and postural transitions. In some cases, this approach can have underestimated the complexity of motor alterations and should be avoided in the future in favor of implementing different kind of instrumental evaluations and clinical scales.

Our study has some limitations. First, the keywords that we used for this research could be, in some cases, slightly different from medical terms or definitions used in the past, especially in very old literature. In order to avoid a problem deriving from a large number of different expressions and definitions, we decided to rigorously adopt only MeSH Terms. This may have led to a potential loss of data. Second, our research on the literature did not find any previous systematic review but only case reports/series, book chapters/lectures and very heterogenous experimental studies. For this reason, instead of a systematic approach, we decided to organize the review in a chronological order. This methodology may have partially affected an appropriate presentation of relevant data.

## 6. Conclusions

We believe that our review can contribute to shedding light on the definition of gait apraxia in iNPH. It also raises questions that remain unanswered, as follows. Is it correct to define apraxia of gait as a walking disorder characterized by stereotyped and perseverating motor behaviors that interfere with the modulation of gait pattern? How can this perseveration phenomenon affect postural transitions? Would it be better to define motor alterations in iNPH as “Gait and Postural Transitions Apraxia”? What is the role of the supplementary motor area? Further studies are needed to improve our knowledge about the pathophysiology of motor alterations in Hakim’s disease. Perhaps Fisher was right when he ironically wrote: “*It is obviously easier to say what the gait disturbance is not than what it is*”. After 45 years, this statement essentially still applies.

## Figures and Tables

**Figure 1 biomedicines-11-01086-f001:**
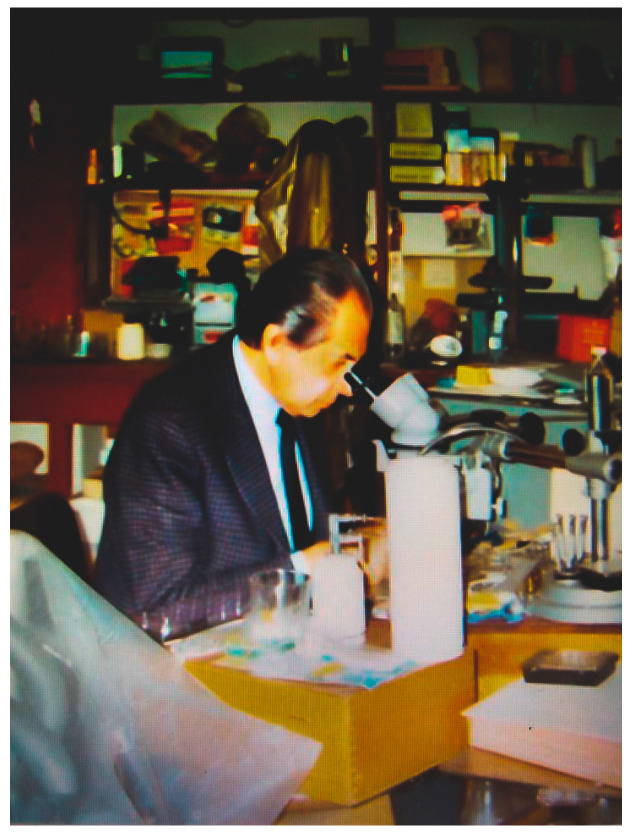
Prof. Salomón Hakim.

**Figure 2 biomedicines-11-01086-f002:**
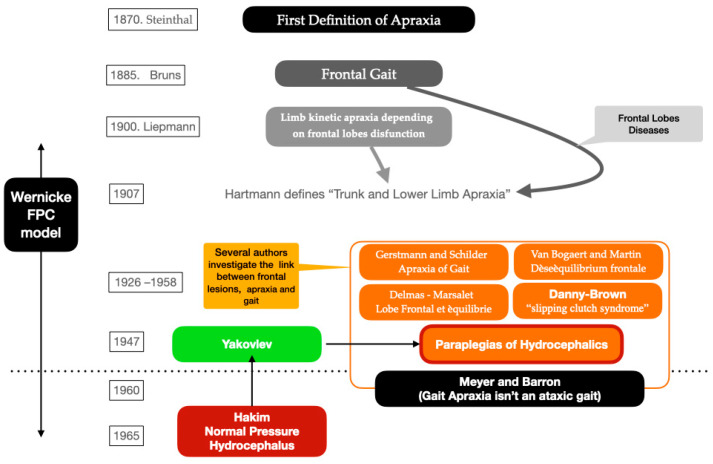
The historical evolution of the concept of gait apraxia in frontal diseases.

**Figure 3 biomedicines-11-01086-f003:**
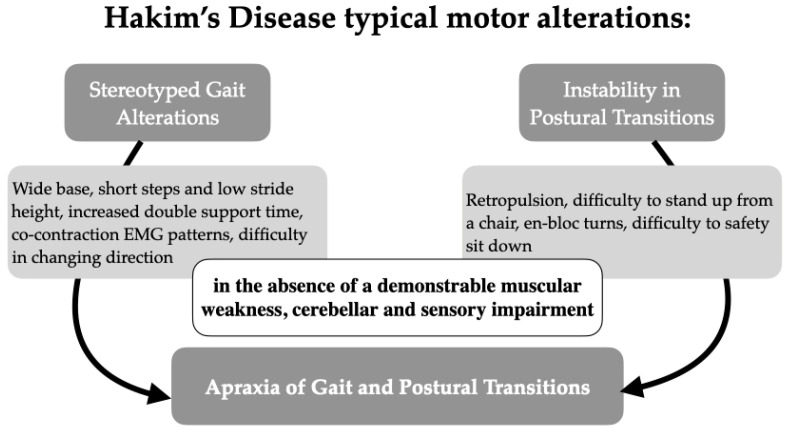
Proposal for a definition of motor abnormalities in idiopathic normal pressure hydrocephalus.

**Table 1 biomedicines-11-01086-t001:** Type of publications included in the final review.

Article Type	Number of Articles	Notes
Case Report/Series	14	
Experimental Studies	8	Very heterogeneous design, no relevant RCT were found in the literature.
Reviews	8	No relevant systematic reviews were found in the literature
Book Chapters	2	

## Data Availability

The authors declare that no new data were created for this narrative review.

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
