# Peer review of "Gait Apraxia and Hakim’s Disease: A Historical Review"

_biomedicines, 2023, doi:10.3390/biomedicines11041086_

Round 1
Reviewer 1 Report
Thank you for the possibility of reviewing this interesting manuscript. Manuscript title: Gait Apraxia and Hakim’s Disease. Generally, the authors partition the manuscript into two main parts.
In this historical review authors describe the origins of the terms “Gait Apraxia”, “Frontal Gait” and “Bruns’ Ataxia”, starting from the first works of Carl Maria Finkelburg, Fritsch and Hitzig and Steinthal during the second half of the 19th century, up to Hakim studies and his formal definition of idiopathic Normal Pressure Hydrocephalus (iNPH).
In the second part of the review, authors analyze how and why these definitions of gait have been associated to Hakim’s disease, in the following literature from 1965 up to nowadays. The definition of “Gait and Postural Transition Apraxia” is then proposed, but fundamental questions about the nature and mechanisms underlying this condition remain unanswered.
In my opinion, the article does not meet the requirements types of
publications. Reviews offer a comprehensive analysis of the existing literature within a field of study, identifying current gaps or problems. They should be critical and constructive and provide recommendations for future research.
1. The title should be changed and made more specific, at the moment it is very general and sounds like a chapter in a book, not the topic of a scientific article.
2. The Introduction should contain a theoretical description. Based on a review of existing scientific reports. For this purpose, a thorough review of the literature is needed, specifying the number of articles related to the topic of the work, a method for selecting articles that match the article's assumptions, objectives and hypotheses. Articles should come from international databases, which should be listed in the Methods section.
3. Then should be presented literature within a field of study, identifying current gaps or problems.
4. The next part of the article should clearly describe future research and authors' recommendations.
Reviewer 2 Report
This manuscript entitled “Gait Apraxia and Hakim’s Disease” was primarily aimed to historically review the “Frontal Gait”, “Bruns’ Ataxia” and “Gait Apraxia”. The authors bring an interesting study, but there are still some problems that cannot up this article to a publishing level. Suggestions are listed in the specific comments below.
Specific comments:
1. In the abstract part, line 16, “we describe the origins of the terms “Gait Apraxia”, “Frontal Gait” …” please write it in the past tense.
2. In the abstract part, line 19, “we analyze how…” please write it in the past tense.
3. In the introduction part, line 29-31, “In recent years Gait Analysis (GA) has progressively led to a renewed interest of re-searchers and physicians on gait alterations of idiopathic normal pressure hydrocephalus (iNPH).” Please cite relevant papers here.
4. In the part 1 and part 2, from my point of view, it is recommended to delete figure 1 and 2 which are not relevant to the content of the article.
5. In the part 2, line 46-48, “This theory found great approval by his mentor Wernicke who hypothesized a damage in frontopontocerebellar fibers as responsible for such disturbances.” Please cite the relevant paper here.
6. In the part 3, line 106, please replace “knee” with “knees”.
7. In the part 3, many statements lack reference, please check and cite the relevant papers.
8. In the figure 3, please replace “he” with “the”, which is probably a typo. Besides, the pixel in Figure 3 is too blurred, please replace a clearer figure.
9. In the discussion part, line 227-228, “Two main parallel theories have developed after Liepmann’s formal definition of apraxia [9].” Since there are two main parallel theories, one more article should also be cited here. Some recently studies could be added in the discussion, such as:
Molecular Characterization of Portuguese Patients with Hereditary Cerebellar Ataxia. Cells 2022, 11, 981. https://doi.org/10.3390/cells11060981
Foot Morphology and Running Gait Pattern between the Left and Right Limbs in Recreational Runners. Physical Activity and Health, 7(1), 43–52. DOI: http://doi.org/10.5334/paah.226
10. In the discussion part, is the Figure 4 put forward by the authors? If so, please also provide detailed descriptions in the article.
11. In the conclusion part, please provide relevant descriptions about the implications for the future studies.
12. Please do check the language and grammar mistakes throughout the whole article to further improve clarity.
Round 2
Reviewer 1 Report
The Authors significantly revised the manuscript. I am satisfied.
Reviewer 2 Report
All my questions have been well addressed, I recommend to accept now.